# Factors influencing participation in randomised clinical trials among patients with early Barrett's neoplasia: a multicentre interview study

Mohammad Farhad Peerally ,[1,2] Clare Jackson,[3] Pradeep Bhandari,[4] Krish Ragunath ,[5,6] Hugh Barr,[7] Clive Stokes,[8] Rehan Haidry,[9] Laurence B Lovat,[10] Howard Smart,[11] John De Caestecker[12]

For numbered affiliations see end of article.

**Correspondence to**
Dr Mohammad Farhad Peerally;
mfp6@le.ac.uk

## ABSTRACT

**Objectives** Strong recruitment and retention into randomised controlled trials involving invasive therapies is a matter of priority to ensure better achievement of trial aims. The BRIDE (Barrett's Randomised Intervention for Dysplasia by Endoscopy) Study investigated the feasibility of undertaking a multicentre randomised controlled trial comparing argon plasma coagulation and radiofrequency ablation, following endoscopic resection, for the management of early Barrett's neoplasia. This paper aims to identify factors influencing patients' participation in the BRIDE Study and determine their views regarding acceptability of a potential future trial comparing surgery with endotherapy.

**Design** A semistructured telephone interview study was performed, including both patients who accepted and declined to participate in the BRIDE trial. Interview data were analysed using the constant comparison approach to identify recurring themes.

**Setting** Interview participants were recruited from across six UK tertiary centres where the BRIDE trial was conducted.

**Participants** We interviewed 18 participants, including 11 participants in the BRIDE trial and 7 who declined.

**Results** Four themes were identified centred around interviewees' decision to accept or decline participation in the BRIDE trial and a potential future trial comparing endotherapy with surgery: (1) influence of the recruitment process and participant–recruiter relationship; (2) participants' views of the design and aim of the study; (3) conditional altruism as a determining factor and (4) participants' perceptions of surgical risks versus less invasive treatments.

**Conclusion** We identified four main influences to optimising recruitment and retention to a randomised controlled trial comparing endotherapies in patients with early Barrett's-related neoplasia. These findings highlight the importance of qualitative research to inform the design of larger randomised controlled trials.

## BACKGROUND

Randomised controlled trials (RCTs) are widely recognised as the most robust way of determining cause and effect between

## STRENGTHS AND LIMITATIONS OF THIS STUDY

⇒ This qualitative study synthesises evidence from both patients who accepted to participate and those who rejected participation in a feasibility randomised controlled trial (RCT), allowing a broader understanding of factors influencing participation in RCTs involving endoscopic procedures for the management of early Barrett's-related neoplasia and those comparing endoscopy with surgery.

⇒ Participants were recruited from all six participating sites in the BRIDE trial, maximising transferability of results.

⇒ The interviews were conducted by telephone, allowing interviewees to participate from the comfort of their own familiar environments.

⇒ The conduct of interviews by telephone also limited the ability of the interviewer to detect non-visual cues from the participants.

⇒ The study was conducted before the COVID-19 pandemic. Different themes may have emerged from repeating the study after the pandemic.

treatment and outcomes.[1] To ensure statistical power and validity of results, strong recruitment and retention into RCTs is a matter of priority for investigators, clinicians and funders. A review of trials funded by two of the UK's biggest funding agencies (the UK Medical Research Council and the National Institute of Health Research Health Technology Assessment Programme) found that less than one-third of trials achieved their recruitment targets.[2] Two Cochrane reviews investigating influences to patient participation in RCTs have identified some effective interventions, such as telephone reminders to non-respondents, use of opt-out instead of opt-in procedures for initiating patient contact and open trial designs, but many such strategies may also pose ethical issues.[3]

Factors affecting the recruitment and retention of participants into research involving endoscopy interventions have not been previously investigated. Yet, there is a need for adequately powered RCTs to determine both safety and effectiveness of such invasive procedures when comparing different types of endotherapies with each other and with more invasive surgical alternatives. Studies using qualitative methods can investigate issues relating to trial recruitment which may not be captured using quantitative methods, by exploring the experiences of participants and recruiters.[4] One area of gastrointestinal medicine where therapeutic options include both endotherapies and surgery is the management of early oesophageal cancer secondary to Barrett's oesophagus (BO). BO is a premalignant condition where the stratified squamous epithelial lining of the distal oesophagus is replaced with a pathological, specialised columnar epithelium. It occurs as a consequence of chronic inflammation from gastro-oesophageal reflux disease.[5] It may further progress through degrees of dysplasia to adenocarcinoma.[6] In the presence of high-grade dysplasia (HGD), BO carries a high risk of progression[7 8] which, if diagnosed once symptoms have occurred, has a very poor prognosis with 5-year survival rates of around 10%–15% in England and Wales.[9] If HGD or intramucosal cancer is identified, macroscopically visible lesions are usually removed by endoscopic resection (ER) but it may leave behind BO where precancerous lesions (dysplasia) can recur.[10]

The aim of the BRIDE (Barrett's Randomised Intervention for Dysplasia by Endoscopy) Study was to investigate the feasibility of undertaking a multicentre RCT comparing two ablative endotherapies used in addition to ER in the management of early Barrett's-related neoplasia (HGD or T1 adenocarcinoma): argon plasma coagulation (APC) and radiofrequency ablation (RFA), both of which had shown promising results in clinical trials where they were compared for post-ER mucosal ablation with surveillance[11] and sham.[12] This paper describes factors identified from semistructured interviews of patients approached to participate in BRIDE, influencing their decision to accept or decline to participate, and to determine their views regarding acceptability of a potential future trial comparing surgery with endotherapy.

## METHODS

This qualitative study consisted of semistructured telephone interviews with a purposive sample of patients contacted for participation in BRIDE. The sampling strategy was designed to include a wide range of views and experiences, including those who agreed to be part of the trial and those who did not, from across all six participating English centres. Potential participants were approached for recruitment into the clinical trial and the interview study when they attended for either their outpatient appointment or their endoscopy, after being identified at a local cancer multidisciplinary team (MDT) meeting. They were all asked whether they would be willing to participate in the interview study, irrespective of their decision to enlist on the trial. Potential participants were given a minimum of 48 hours to consider the information given to them about enrolment into the trial and/or interview study. More details on the patient identification process from the six participating centres have been described previously.[13] They were chosen to represent a broad range of characteristics based on age, sex, centre and decision to participate or not participate in the trial. Participants who were interested in taking part in the interview study were sent an information sheet by post, with return envelopes for signed consent forms and contact details for the telephone interview.

To explore the feasibility of conducting larger fully powered trials, the interviews focused on views and experiences of being invited to participate in BRIDE and participation in a potential future trial of endotherapy versus surgery. Interviews were conducted by an experienced non-clinical qualitative researcher, who was not involved in the recruitment of patients in the trial. They occurred in the first year after patients were approached, and were structured flexibly around an interview topic guide, developed in an inductive manner by the authors and informed by discussions with lay representatives (online supplemental material A). No further participants were recruited after achieving theoretical saturation.[14]

All tapes were transcribed verbatim and anonymised. Analysis was performed by author CJ, a qualitative researcher, based on the constant comparative method, which comprised a systematic approach to data analysis involving (1) open coding of interview transcripts, while comparing codes across transcripts, (2) axial coding where the interplay between codes was explored to create categories connecting codes together and (3) selective coding of categories to create higher-order thematic categories.[15] Data analysis was managed by QSR V.N6 software. To further increase trustworthiness of the data, investigator triangulation[16] was performed with two other authors (MFP and JDC) reviewing the interpretations of the findings against the participants' quotations.

### Patient and public involvement

The interview topic guide was developed collaboratively with two members of the public with lived experience of BO and oesophageal cancer, who virtually commented on a draft version of the topic guide.

## RESULTS

We interviewed 18 participants: 16 men and 2 women, aged 47–85 years from all six treatment centres. Eleven interviewees participated in the BRIDE trial and seven chose not to. Interviews lasted between 26 and 79 min. When discussing factors influencing their decisions whether or not to enrol into the BRIDE trial and a potential future trial comparing endotherapy with surgery, participants mentioned a number of considerations, which have been synthesised into four main themes describing factors

intrinsic to: (1) the nature of the trial recruitment process along with the participants' relationship with the trial recruiters and clinical staff, (2) the participants' perceptions of the design and aim of the study, (3) participants' motivations for participation balanced against the burden of participation (conditional altruism), and (4) participants' perceptions of risks of surgery versus endotherapies. These themes are described in more details below.

### Nature of recruitment process and role of recruiters

Participants valued the interpersonal skills of the recruiters, irrespective of their grades or professions, placing emphasis on their attitudes and respectful approach. Such attributes translated to a smooth recruitment process to the trial, thereby ensuring that participants did not feel any pressure to participate.

> I mean she [the recruiter] was very pleasant …and I think if you've got a disposition like that…it puts you at ease … there's no way that I thought to myself 'I'm pressurised into it' and I fully understood that if ever I wanted to pull out of the study then I can. (Participant 12)

> …it's a very sort of informal situation which I was very comfortable with. I think I'd find it more difficult if I had some very formal doctor. I think the relationship I just feel was being, right from the first I found it very easy to talk to and they were very open in all aspects and they were very inclusive of the patient down there. (Participant 14)

Participants also placed considerable trust in their clinical team, whom they viewed as the experts in managing their condition. A product of this trust was the belief that, if their clinical team was discussing the option of enrolling into a particular trial, it had to be beneficial for them. Such trust lessened the weight participants placed on written information provided to them as part of the recruitment process, and facilitated their decision-making.

> …if he [the consultant] said that he reckoned he could improve the situation, then I was quite happy at just letting him get on with it. (Participant 15)

> I'd already had some treatments from [Doctor A], and I'd got every confidence in him, so when he came along and said, 'We want to do this and we want to do that', I'd every confidence in him. (Participant 13)

Our findings illustrate the importance of adequate planning when approaching patients for recruitment into trials. On two occasions, participants received information about the trial before being told that they had early cancer, creating significant distress leading to one of the participants declining to be part of the trial. Approaches seen as unconventional by patients, such as being called to a different room or their clinician asking for a nurse or another clinician to be present, caused concern to patients.

> The first time I heard about [the trial] was when I received information through the post… But I hadn't even had my results back from the hospital. Nobody had explained to me what Barrett's Oesophagus was…so to then receive the package, because it was quite a substantial package with consent forms and everything…I wasn't contacted after I had my results or sat down with anybody, I was contacted beforehand and that was a bit alarming…Not just for myself …it was quite upsetting for my wife and my daughter. (Participant 6)

> Before we went to see the surgeon, he [the researcher] wrote a letter to us saying that 'oh you've been diagnosed with this cancer' and he wanted to study, well we didn't know about it… they said 'oh, [Dr B] wants to see you… and your wife'… we went to see him… and the wife says to him 'OK, if it's bad news, give us the bad news', 'oh no, no', because he'd called in another consultant as well, he called in another consultant and we were wondering… why is he calling another consultant [trial investigator] in, I thought that this meant bad… (Participant 18)

### Participants' perceptions of the study

Participants' understanding of the purpose of the trial varied widely. Participants conflated the aims of the *feasibility* study which they were enrolling in (which was to investigate whether it was possible to conduct a more definitive adequately powered future study) with that of a *definitive* study itself. Those who agreed to participate largely understood that available evidence at the time of the trial showed that both interventions were effective. This belief in clinical equipoise was essential to participants' acceptance for enrolling into the trial and be randomised to one intervention or the other.

> People might be a bit reluctant if they think there's a 50/50 chance of them getting a treatment that's more painful or hard to go through, whereas if both options were pretty much the same, which I believe is so in my case, I've got to have treatment so it's a flip of the coin which I get, I don't really mind. (Participant 9)

> Reading on the Internet what it was about, the two procedures and I just got it in my mind that there were roughly the same treatment… (Participant 12)

After making their own research into the two interventions, one interviewee did not believe that both treatments were equal, and thereby declined to be part of the trial, choosing instead to receive, what they believed, was the more effective treatment.

> I'd done some research…my wife had looked into it as well on the treatment and I couldn't be guaranteed that I'd be given Halo [RFA] treatment. The Halo treatment was a little bit more successful than one of the other treatments that was being

offered. So, I then withdrew my consent at the time… (Participant 6)

The study design was not always understood by participants, with some believing that their treatment modality was individualised to their needs as opposed to being chosen at random. Such therapeutic misconceptions[17] demonstrated that participants did not always understand the implication of their decisions to enrol in the study and impinged on participants' autonomous decision-making and their ability to give informed consent.

No, he [the clinician] had decided that the gas [APC]… would be the best for me… I was quite happy to go along with whatever he suggested. (Participant 15)

Alarmingly, one participant who had been recruited to take part in the study did not appear to have any idea what the study was about.

What was the study about, did you understand what was going to go on? [to someone who appeared to be the participant's partner]. … I can't answer your question, I don't know. (Participant 7)

The implication of non-participation in the trial was also not uniformly understood among participants. Two interviewees believed that active treatment would only be offered to them if they agreed to take part in the trial.

I don't think [the doctor] really had a choice. It wasn't a no choice it's a yes go forward because all the choice was just let it carry on and let it take your life isn't it. (Participant 2)

I was just approached to say that there were several possible alternative courses of treatment and one of them might involve the BRIDE [trial] effectively. So, I was asked was I willing to consider any of those and I said 'well, of course, because I want to get better, all being well'. (Participant 17)

### Conditional altruism

When discussing reasons for enrolling in the trial, interviewees widely reported being motivated by the need to help others and contribute to medical advancement.

As I say, it was simply if it helps somebody in the future, good. 'Cause I'm sure somebody [would have] done exactly the same for the treatment I just had. (Participant 4)

A letter came through later on and we just thought that we'd help because the National Health, they did a fantastic job on me and basically they saved my life. (Participant 18)

Such descriptions of altruism and reciprocity were not always unconditional. They were frequently combined with an expectation that participants would also benefit by enrolling in the trial or at least be no worse off. Taking part in the trial was viewed by interviewees as a means of receiving better care by having access to additional information, support and monitoring, which may not be available through standard care.

…I thought maybe I would get monitored more that was my main reason: I really did think that I would be monitored more. (Participant 8)

…I work on the principle that hopefully it will help me and whatever findings come out at the end of the trial hopefully it will help somebody else down the line… (Participant 11)

That's right and being on the study, I think you get far better treatment as well. (Participant 16)

As well as considering personal and altruistic motives for participating in the trial, interviewees also gave significance to the amount of personal inconvenience that participating in the trial might entail. Practical concerns included travel, inconvenience to themselves or family, health and financial considerations. Such considerations were largely dependent on the individual's personal circumstances at the time of the approach.

The main trouble with this study is that it's in [Place 2] and I live in [Place 3], so it's a 40-odd mile trip down. And obviously, having had anaesthetic, I can't drive back. And so one of my daughters had taken me down. (Participant 13)

The trouble was at the time my late husband was very ill and so it was a question that you know, it was the time factor you know, that I was, I mean OK I wanted to keep myself going for him but also I had to look after him so I couldn't sort of give my full attention to it. (Participant 11)

### Risks of surgery

Taking part in the trial was also subject to there being no significant extra treatment or risk to themselves in taking part. Such reflections on risk were fuelled by the fact that both interventions were endotherapies, considered to have similar risk profiles, unlike more invasive surgery.

…I don't think I had anything to worry about, it [either endotherapy] wasn't going to hurt me or set me back or done any damage to me, it wasn't a risk that way really…they're trying to save you the trauma of a big operation. (Participant 18)

I think as I've already said, as long as I haven't got to have any extra treatment and it's not going to be much trouble. (Participant 9)

When presented with information about a potential future trial comparing surgery with endotherapy for the management of the same condition, interviewees recognised the differences in the two interventions much more readily. They were not seen as broadly equivalent. Surgery, for many, was seen as riskier and involved additional suffering alongside a longer recovery time. Therefore,

participants rejected the notion of clinical equipoise in such a potential future trial.

> I think really if it was going to be a flip of a coin, I think that's no way to treat a patient is it. (Participant 7)

> …anything that can be done with an endoscopy must be better than cutting people open. Would have to be at a much greater, later stage and need for people to be opened up. (Participant 8)

In order to participate in a trial which might involve surgery as intervention when compared with endotherapy, participants felt they would require more information on efficacy and safety. Ironically, both of these determinants would not be known before the trial, since a trial comparing surgery with endotherapy in the management of early Barrett's-related oesophageal cancer would seek to address these exact answers.

> Well, if they [researchers] said either is possible, they'd [participants] want to know which one is the most successful. (Participant 4)

> It depends why they [the researchers] want to do it and what are the advantages of having a surgical procedure as opposed to an endoscopy… But I wouldn't say I wouldn't have it, I would want to know a hell of a lot more about it and what the pros and cons are. (Participant 1)

## DISCUSSION

This study highlights important patient-centred considerations when designing and conducting trials involving endotherapies for early oesophageal cancer. Our findings

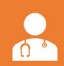 Better coordination between clinical and research teams before contacting patients for trial participation

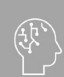 Simplify technical terms related to trial design (e.g. randomisation, equipoise)

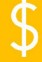 Consider monetary incentives for participants

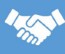 Share good practice between recruitment centres

**Figure 1** Practical considerations to maximise recruitment into randomised controlled trials comparing different endotherapies for the management of patients with early oesophageal cancer.

demonstrate the interactions of recruiters with potential participants before recruitment played a key role in influencing recruitment. Participants valued the respectful and pleasant nature of these interactions, particularly when they were not placed under any duress to participate. Recruitment was facilitated by the dual role played by recruiters who were also often the participants' trusted clinicians. This finding mirrors the results of other interview studies of trial recruiters.[18 19]

Being both a patient's clinician and a trial researcher can be associated with certain emotional conflicts for the clinical researcher, which need to be acknowledged.[18 20] For instance, in an interview study, Donovan *et al* highlight challenges expressed by clinicians recruiting into trials around their own treatment preferences,[18] based on their personal opinions. Equally, patients' trust in their clinicians should not be abused to facilitate recruitment. Recruiting clinicians have a duty to be honest and convey both the genuine uncertainty between the safety and effectiveness of the different interventions in the trial (ie, the clinical equipoise) and be clear on the purpose of feasibility trials, which are designed to inform the design of future more definitive trials. We found both concepts to be poorly understood by certain participants, highlighting potential gaps in the explanation conveyed to them. The literature is also lacking on how recruiters describe and operationalise the concept of equipoise during RCT recruitment.[18 21]

Our findings highlight the importance of judiciously timed discussions regarding enrolment into studies, in particular where breaking bad news may be involved. We identified instances when patients were contacted to enrol in research before being told of the diagnosis and management options. We suspect that this situation arose because potential participants for the trial were identified through cancer MDTs by researchers who were not part of their usual clinical teams. While trial recommendation through MDTs has previously been shown to increase recruitment rates,[22] our findings suggest that researchers should not fall into the trap of 'recruitment enthusiasm' and ensure that patients are aware of their diagnoses before contact is made. Unexpected deviations from usual outpatient practices when recruiting patients into trials (eg, having two consultants seeing a patient) also created undue distress.

While altruism did play a factor in influencing recruitment, we found that participants also considered the degree of inconvenience trial participation would entail. McCann *et al* previously coined the term 'conditional altruism' to describe participants' willingness to participate in trials to contribute to a greater good so long as they reap some personal benefit from the trial or at least not get harmed.[23] We find that such conditional altruism also entailed an expectation that participation would not be accompanied by personal inconvenience. This finding is particularly relevant for interventions delivered in tertiary centres, as is recommended for the endoscopic management of patients with Barrett's-related

neoplasia,[24] and which may involve participants' time and financial sacrifice, as was the case for the BRIDE trial.

Our findings would suggest that surgery would not be acceptable for participants invited to participate in a trial comparing surgery with endotherapy for early oesophageal cancer unless there was a potential advantage to the more invasive intervention (ie, surgery) such as higher likelihood of cure, counterbalancing the downsides of increased likelihood of complications. A recent quality of life survey comparing patients with neoplastic BO who had undergone endotherapy with those who had oesophagectomy supports this view with the latter group suffering from more long-term symptomatic burden.[25] The issue of recruitment into trials involving surgical interventions is not new, with multiple previous reviews highlighting numerous barriers to such RCTs: the irreversible nature of surgery, strong patient treatment preferences, surgeons' personal opinions and concerns around the idea of randomisation.[26–28] We suggest that surgery is only considered an option when compared with endotherapy when there is an established evidence base for the surgical intervention such as high likelihood of cure, counterbalancing the risks of complications. For instance, early oesophageal cancer is associated with certain histological prognostic factors which increase the likelihood of lymphatic spread (such as submucosal invasion to more than 500 μm, or lymphovascular invasion or poorly differentiated tumours).[29 30] In such cases, surgery would offer a higher likelihood of cure since local lymph nodes are also removed with the oesophageal resection.[31]

Based on our findings, we suggest some practical considerations to maximising recruitment into future RCTs comparing different endotherapies for the management of patients with early oesophageal cancer. These recommendations may be generally applicable to other RCTs involving invasive interventions, and are summarised in figure 1.

First, there is a need for better coordination between clinical and research teams when contacting potential participants for enrolment. If participants are identified through MDTs, they should only be contacted for participation after being made aware of their diagnoses and treatment options by their clinicians. Clearly, such coordination is made smoother if their own clinician is the trial researcher.

Second, trial recruiters need to acknowledge difficulties in explaining complex concepts such as equipoise and randomisation to patients. Previous trials have used peer feedback to recruiters to improve their own communication and enhance patient written communication including descriptions of complex trial concepts and interventions in layman's terms.[32]

Third, monetary incentives,[33] at the very least, to cover subsistence expenses need to be considered. Such incentives are particularly necessary when patients and carers are expected to cover long distances for treatment as part of a trial.

Fourth, we suggest that multicentre trials need to include opportunities for peer feedback between researchers from different centres during the conduct of a trial to allow sharing of good practice promoting recruitment and discuss challenges.

This study is limited by the small sample although the qualitative methodology used means that the findings are still valid given that sample size in such research is guided by theoretical saturation.[15] Nonetheless, a bigger sample may have led to more themes, as would have inclusion of trial researchers among the interviewees. The study was also conducted before the COVID-19 pandemic. Given the effect of the COVID-19 pandemic on patients' lifestyles and access to care, other themes may have been generated from repeating the study during or after the pandemic. Finally, to allow time for all participants to complete their treatment plans as per trial protocols, interviews were conducted up to a year after participants were initially approached. Such a timescale may have introduced recall bias to the findings of this study.

In conclusion, recruitment and retention of participants into RCTs involving endotherapies is not straightforward. We have demonstrated the value of conducting qualitative research as part of feasibility trials to inform the design of larger RCTs. We highlight both opportunities and challenges in maximising recruitment to future trials comparing endotherapies and endotherapy versus surgery for the management of patients with early oesophageal cancer.

Based on our findings, we have made some practical recommendations to optimising recruitment and retention in RCTs involving invasive treatments (both endotherapy and/or surgery) for the management of early BO-related neoplasia. These recommendations may be generally applicable to other RCTs involving invasive interventions. Future research should aim to evaluate the views of researchers and clinicians on factors influencing participation in trials comparing different endotherapies for the management of BO-related early oesophageal cancer and the effect of the suggested recommendations on recruitment.

**Author affiliations**
[1]SAPPHIRE, Department of Health Sciences, University of Leicester, Leicester, UK
[2]Digestive Diseases Unit, Kettering General Hospital, Kettering, UK
[3]Faculty of Medicine and Health Sciences, University of Nottingham, Nottingham, UK
[4]Queen Alexandra Hospital, Portsmouth, UK
[5]Department of Gastroenterology & Hepatology, Royal Perth Hospital, Perth, Western Australia, Australia
[6]Faculty of Health Sciences, Curtin Medical School, Perth, Western Australia, Australia
[7]Department of Surgery, Gloucestershire Royal Hospital, Gloucester, UK
[8]Chestnut House, Gloucester Royal Hospital, Gloucester, UK
[9]Department of Gastroenterology and Hepatology, University College Hospital, London, UK
[10]Division of Surgery and Interventional Science, University College London, London, UK
[11]Department of Gastroenterology, Royal Liverpool Hospital, Liverpool, UK
[12]Digestive Diseases Centre, University Hospitals of Leicester NHS Trust, Leicester, UK

**Acknowledgements** We would like to thank our two expert patient advisers for their contribution to the design of the study.

**Contributors** MFP wrote the initial manuscript, edited subsequent versions following feedback from all authors and edited the final version of the manuscript. CJ performed the analysis of the data and provided a data summary. CJ, PB, KR, HB, CS, RH, LBL and HS critically revised and edited all versions. JDC reviewed all versions of the manuscript, edited the final version of the manuscript and was responsible for the overall content of the study as guarantor. All authors approved the final manuscript version being submitted for publication.

**Funding** This article presents independent research funded by the National Institute for Health Research (NIHR) under its Research for Patient Benefit Programme (grant no. PB-PG-0711-25066).

**Disclaimer** The views expressed are those of the authors and not necessarily those of the NHS, the NIHR or the Department of Health.

**Competing interests** KR received educational grants from Erbe and Medtronic. RH received research infrastructure support from Medtronic, Cook Endoscopy and Pentax Europe. LBL received research infrastructure support from Medtronic. All other authors disclosed no financial relationships relevant to this article.

**Patient and public involvement** Patients and/or the public were involved in the design, or conduct, or reporting, or dissemination plans of this research. Refer to the Methods section for further details.

**Patient consent for publication** Not required.

**Ethics approval** This study involves human participants and was approved by the Leicester Central National Research Ethics Service Committee, East Midlands Research Ethics Committee (12/EM/0445). Participants gave informed consent to participate in the study before taking part.

**Provenance and peer review** Not commissioned; externally peer reviewed.

**Data availability statement** Data are available upon reasonable request. Reasonable request for data sharing will be considered.

**ORCID iDs**
Mohammad Farhad Peerally http://orcid.org/0000-0003-3004-375X
Krish Ragunath http://orcid.org/0000-0001-6571-5435

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
