## [Reviewer comments · BMJ Open]

ARTICLE DETAILS

TITLE (PROVISIONAL)	Factors influencing participation in randomised clinical trials among patients with Barrett's early neoplasia: A multi-centre interview study
AUTHORS	Peerally, Mohammad; Jackson, Clare; Bhandari, P; Ragonath, Krish; Barr, Hugh; Stokes, Clive; Haidry, Rehan; Lovat, Laurence; Smart, Howard; De Caestecker, John

VERSION 1 – REVIEW

REVIEWER	Sijben, Jasmijn Radboudumc Radboud Institute for Health Sciences
REVIEW RETURNED	17-Jun-2022

GENERAL COMMENTS	General comments to the Author: This well-written paper is very interesting, as it highlights the importance of incorporating individuals' preferences in the design of clinical trials and health care interventions. The relatively high sample proportion of individuals that declined BRIDE participation is a very strong point. However, a number of issues should be addressed before this manuscript can be accepted for publication. Major points: 1. Methods, paragraph 1. Please explain how the decision to end the participant recruitment phase was made. The authors note in the discussion that the findings from this small sample size are still valid, which is true if the point of thematic saturation was reached. It is however not clear from the methods if the authors aimed for thematic saturation. If the authors did not recruit participants until saturation was reached, they should interview more participants.2. Methods, paragraph 2. It is highly recommended to use input from existing literature or a theoretical framework to inform the interview guide.3. Methods, paragraph 3. Please report which author (s) did the analysis. It is recommended to perform data analysis in groups of at least 2 persons.4. Results. Please provide baseline participant characteristics.5. Discussion, paragraph 6-10. I think the recommendations made in the discussion are very important and supported by the data. Is it possible to give these a more prominent location in the manuscript where we can be sure the readership will find them easily? Perhaps add a figure or text box? Minor points: 1. Introduction, paragraph 2. Please state more clearly what the value of qualitative research is in supplementing the knowledge on effective interventions to increase RCT participation that are described in the first paragraph.
--

	2. Introduction, paragraph 2. Please add a reference for the statement about progression of BO through dysplastic stages. 3. Introduction, paragraph 2. Please replace reference 5 and 6 with a reference to the most recent meta-analysis on progression of BO. 4. Methods, paragraph 1. Please elaborate more on the strategy that was used for purposeful sampling. 5. Methods, paragraph 2. Please report more information about the qualitative researcher. What was their professional background? Did he/she/they have a prior relationship with the participants (like a role in the BRIDE trial or a doctor-patient relationship)? 6. Methods, paragraph 4. Why were a patient with esophageal cancer and a carer consulted? In my opinion, their perspectives are not directly related to the perspective of individuals with dysplasia in BO.
--	---

REVIEWER	van Driessche, Anne Vrije Universiteit Brussel Faculteit Geneeskunde en Farmacie
REVIEW RETURNED	19-Sep-2022

GENERAL COMMENTS	Thank you for the opportunity to review this article. The aim of gaining more knowledge on the decisions of participants to either participate or not in a trial such as this one is very relevant for other researchers, and your article includes several important suggestions. The article is of good quality in my opinion. It could use some clarifications on a few aspects:  1. It is mentioned in the methods section that the interview topic guide was developed with lay representatives. In a later stage you mention who these were in the "patient and public involvement section". How did this help in constructing the topic guide? How did you structure this cooperation? (how many times did you meet, how did you include their feedback, did they just review the draft topic list f.e. or were they involved in a more active way?) To what extent would you advise colleagues to do the same? 2. Regarding analysis: you mention the constant comparison method, but could you elaborate more on how you did that? How did you handle any disagreement? Why did you choose that approach for this purpose? 3. In your results section, you don't add information on the division of the participants per recruitment centre. This information could be relevant, as it might be possible that the non-participants are all from 1 centre for example, potentially influencing the interpretation of the results. 4. It is mentioned in the methods section that interviews were conducted in the first year after being approached. A year is quite long. Why did you choose one year? What about the risk of recall bias? Is that mentioned in your discussion? 5. Could you clarify your fourth recommendation you are giving in your discussion: "multi-centre trials need to include opportunities for group feedback during the conduct of a trial to allow researchers to share good practice and discuss challenges". What kind of group do you mean? Do you mean the potential participants themselves, like an advisory group? 6. In your discussion, you mention that coordination is made smoother if their own clinician is the trial researcher. This might not be possible to do in a multi-centre trial. Could you share your experiences in how you tried to stratify the way potential participants were contacted for enrolment?
--

	7. Many references are from before 2015. Not necessarily a problem, but I am just wondering whether in some cases there is a lack of more updated references you could use? Or if not, maybe it could be elaborated in your introduction more clearly why this study is so relevant.
--	--

VERSION 1 – AUTHOR RESPONSE

Reviewer 1

Reviewer 1 comments	Response Comments
This well-written paper is very interesting, as it highlights the importance of incorporating individuals' preferences in the design of clinical trials and health care interventions. The relatively high sample proportion of individuals that declined BRIDE participation is a very strong point. However, a number of issues should be addressed before this manuscript can be accepted for publication.	Thank you for your encouraging comments. We also believe that understanding the factors influencing patient participation in endoscopy RCTs for management of early oesophageal cancer is very important, given the different modalities of endoscopic ablative therapies available, and surgery. We hope to address the points you have raised below.
Methods, paragraph 1. Please explain how the decision to end the participant recruitment phase was made. The authors note in the discussion that the findings from this small sample size are still valid, which is true if the point of thematic saturation was reached. It is however not clear from the methods if the authors aimed for thematic saturation. If the authors did not recruit participants until saturation was reached, they should interview more participants.	As outlined in the discussion section, the sample size was determined by theoretical (or thematic) saturation. This is a commonly used criterion in qualitative research, used to decide when to stop recruitment based on no additional data being found in the analysis. We have made this point clearer in the method section (page 10). "No further participants were recruited after achieving theoretical saturation."¹⁴
Methods, paragraph 2. It is highly recommended to use input from existing literature or a theoretical framework to inform the interview guide.	While we would agree that use of existing literature may have a role when developing interview topic guides, the fact that this study is the first to explore factors influencing patient participation in RCTs for the management of early oesophageal cancer, means that there is no significant relevant literature to inform the design of the interview topic guide.

	Instead, we take a very inductive approach informed by discussions with our lay representatives (as outlined in supplementary material A), the clinical experience of the authors and the methodological expertise of the qualitative researchers in the research team. We have clarified how the topic guide was developed with the following edit – Methods –page 10: “...an interview topic guide, developed in an inductive manner by the authors, and informed by discussions with lay representatives.”
Methods, paragraph 3. Please report which author (s) did the analysis. It is recommended to perform data analysis in groups of at least 2 persons.	We have now included the name of the author who performed the data analysis – page 10, Methods section. “Analysis was performed by author CJ, a qualitative researcher” The concepts of “replication” and “reliability”, which could be maximised by having two or more researchers involved in data analysis, are much more relevant to quantitative research (Maher et al. 2018). Maher, C., Hadfield, M., Hutchings, M., & de Eyto, A. (2018). Ensuring Rigor in Qualitative Data Analysis: A Design Research Approach to Coding Combining NVivo With Traditional Material Methods. International Journal of Qualitative Methods, 17(1). With our piece of research being qualitative in nature, we instead took a number of steps to assure the trustworthiness of the data analysis, in line with norms of practice for qualitative research. Data was coded according to a coding rulebook devised after data immersion, and further enriched with ongoing engagement with the interview transcripts. The codes were then re-applied to the whole dataset. Investigator triangulation was also performed with two other authors.

	We have now added the following line on pages 10-11 to address the reviewer's comment: “Analysis was performed by author CJ, a qualitative researcher, based on the constant comparative method, which comprised a systematic approach to data analysis involving (1) open coding of interview transcripts, while comparing codes across transcripts, (2) axial coding where the interplay between codes was explored to create categories connecting codes together and (3) selective coding of categories to create higher order thematic categories.15 Data analysis was managed by QSR N6 software. To further increase trustworthiness of the data, investigator triangulation16 was performed with two other authors (MFP and JDC) reviewing the interpretations of the findings against the participants’ quotations.”
Results. Please provide baseline participant characteristics.	Baseline characteristics of all included patients have been previously reported in a publication related to the trial findings, which we reference in the manuscript (Ref 13) In this qualitative piece, we only collected age and gender as baseline characteristics (Results section – page 10). The other baseline characteristics (Length of Barrett's, Body Mass Index and Histology) collected as part of the trial and reported in a previous publication (Ref 13 in the manuscript) were not relevant to the outcomes of this particular qualitative paper. They have therefore not been included.
Discussion, paragraph 6-10. I think the recommendations made in the discussion are very important and supported by the data. Is it possible to give these a more prominent location in the manuscript where we can be sure the readership will find them easily? Perhaps add a figure or text box?	We have now summarised the recommendations into four main points in a figure (fig 1).
Introduction, paragraph 2. Please state more clearly what the value of qualitative research is in supplementing the knowledge on effective interventions to increase RCT participation that are described in the first paragraph.	The following line has been added to para 2 in the Background section to address this point: “Studies using qualitative methods can investigate issues relating to trial recruitment which may not be captured using quantitative methods, by exploring the experiences of participants and recruiters.”

Introduction, paragraph 2. Please add a reference for the statement about progression of BO through dysplastic stages.	The following reference has been added (reference 6): Wani S, Falk G, Hall M, et al. Patients with nondysplastic Barrett's esophagus have low risks for developing dysplasia or esophageal adenocarcinoma. Clinical gastroenterology and hepatology 2011;9(3):220-227. e1.
Introduction, paragraph 2. Please replace reference 5 and 6 with a reference to the most recent meta-analysis on progression of BO.	To our knowledge, the latest available meta-analysis specifically looking at the progression from HGD to oesophageal adenocarcinoma is from 2008. The more recent meta-analyses have aimed to look at factors affecting risk of progression, low grade dysplasia, surveillance and ablative therapies. Page 9 – line 2: We have removed the reference from 1983 and included the most recent relevant meta-analysis (reference 8)
Methods, paragraph 1. Please elaborate more on the strategy that was used for purposeful sampling.	Page 9 – methods section. The following line has been added to address this comment: “The sampling strategy was designed to include a wide range of views and experiences, including those who agreed to be part of the trial and those who did not, from across all six participating English centres.”
Methods, paragraph 2. Please report more information about the qualitative researcher. What was their professional background? Did he/she/they have a prior relationship with the participants (like a role in the BRIDE trial or a doctor-patient relationship)?	Page 10, para 2. The following line has been added: “Interviews were conducted by an experienced non-clinical qualitative researcher, who was not involved in the recruitment of patients in the trial.”
Methods, paragraph 4. Why were a patient with esophageal cancer and a carer consulted? In my opinion, their perspectives are not directly related to the perspective of individuals with dysplasia in BO.	Due to the high risk of progression from high-grade dysplasia to oesophageal cancer, patients with high-grade dysplasia are treated similarly as those with early oesophageal cancer according to Barrett’s guidelines in the UK and the US. Thus, we strongly believe that the perspective of patients with HGD are equally as relevant as those with early oesophageal cancer. Co-design of research procedures with lay participants is highly valued in NIHR funded research and is viewed as good practice by funders and policy makers.

Reviewer 2 comments	Response Comments
Thank you for the opportunity to review this article. The aim of gaining more knowledge on the decisions of participants to either participate or not in a trial such as this one is very relevant for other researchers, and your article includes several important suggestions. The article is of good quality in my opinion. It could use some clarifications on a few aspects:	Thank you for your very encouraging comment. We certainly also agree that lessons learnt from this piece of research has wider applicability.
It is mentioned in the methods section that the interview topic guide was developed with lay representatives. In a later stage you mention who these were in the "patient and public involvement section". How did this help in constructing the topic guide? How did you structure this cooperation? (how many times did you meet, how did you include their feedback, did they just review the draft topic list f.e. or were they involved in a more active way?) To what extent would you advise colleagues to do the same?	We had two patient representatives who virtually commented on the topic guide, which was then amended. Co-design of research with patient and public representatives allows meaningful output and explorations of unique perspectives. However, the scope of this study was not particularly set at exploring this particular research practice. Thus, we do not feel it is appropriate for us to further advise researchers on topic guide co-design within the scope of this paper. We have edited the PPI section (page 11) to reflect the above point: “The interview topic guide was developed collaboratively with two members of the public with lived experience of Barrett’s oesophagus and oesophageal cancer, who virtually commented on a draft version of the topic guide.”
Regarding analysis: you mention the constant comparison method, but could you elaborate more on how you did that?	The constant comparison method is a qualitative data analytical technique, with roots in grounded theory. It allows patterns to emerge from data through continuous interaction by the researcher with transcripts already read and analysed, and new transcripts. We have provided more information on the constant comparison technique – Methods page 10: “Analysis was performed by author CJ, a qualitative researcher, based on the constant comparative method, which comprised a systematic approach to data analysis involving (1) open coding of interview transcripts, while comparing codes across transcripts, (2) axial coding where the interplay between codes was explored to create categories connecting

How did you handle any disagreement? Why did you choose that approach for this purpose?	codes together, and (3) selective coding of categories to create higher order thematic categories.” Data triangulation was performed with two other researchers (MFP and JDC) who reviewed interpretations of findings against participants’ quotations, before reaching consensus. The following line has been added to clarify this point – Methods page 11: “To further increase trustworthiness of the data, investigator triangulation was performed with two other authors (MFP and JDC) reviewing the interpretations of the findings against the participants’ quotations.” As an iterative data analytical method, it allows a researcher to analyse data using a systematic approach, increasing trustworthiness of the data. This point is now clarified in the following line – Methods Page 10 “All tapes were transcribed verbatim and anonymized. Analysis was performed by author CJ, based on the constant comparative method, which comprised a systematic approach to data analysis...”
In your results section, you don't add information on the division of the participants per recruitment centre. This information could be relevant, as it might be possible that the non-participants are all from 1 centre for example, potentially influencing the interpretation of the results.	When conducting the interview study, we did not routinely collect information regarding the recruitment site. This information was however collected when the trial was running. Due to anonymisation and de-identification of person identifying data before qualitative analysis, identification of patient at site level is not possible retrospectively. All six participating centres recruited participants using similar procedures after identifying them from the multidisciplinary team meetings. The recruitment procedures have been described in the publication related to the trial results (Reference 13).

It is mentioned in the methods section that interviews were conducted in the first year after being approached. A year is quite long. Why did you choose one year? What about the risk of recall bias? Is that mentioned in your discussion?	A year was chosen to give participants time to complete the treatment plans (ablative endotherapies), scheduled as part of the BRIDE trial, so they could give an informed view regarding the experience of participating in BRIDE to the researcher. We agree that such a timescale introduced recall bias and have included the following statement to elaborate on this point – page 22, Discussion: “Finally, to allow time for all participants to complete their treatment plans as per trial protocol, interviews were conducted up to a year after participants were initially approached. Such a timescale may have introduced recall bias to the findings of this study.”
Could you clarify your fourth recommendation you are giving in your discussion: "multi-centre trials need to include opportunities for group feedback during the conduct of a trial to allow researchers to share good practice and discuss challenges". What kind of group do you mean? Do you mean the potential participants themselves, like an advisory group?	We were specifically referring to peer feedback from researchers of the different centres involved in the trial. We have edited our fourth recommendation as follows (Page 22- Discussion) “Fourth, we suggest that multi-centre trials need to include opportunities for peer feedback between researchers from different centres during the conduct of a trial to allow sharing of good practice promoting recruitment and discuss challenges.”
In your discussion, you mention that coordination is made smoother if their own clinician is the trial researcher. This might not be possible to do in a multi-centre trial. Could you share your experiences in how you tried to stratify the way potential participants were contacted for enrolment?	We have added the following sentence to provide further clarification on how they were contacted in the Methods section – pages 9-10. “Potential participants from all six centres were approached for recruitment into the clinical trial and the interview study when they attended for either their outpatient appointment or their endoscopy, after being identified at a local cancer multi-disciplinary team meeting (MDT). They were all asked whether they would be willing to participate in the interview study, irrespective of their decision to enlist on the trial. Potential participants were given a minimum of 48 hours to consider the information given to them about enrolment into the trial and/or interview study.”
Many references are from before 2015. Not necessarily a problem, but I am just wondering whether in some cases there is a lack of more updated references you could	We have updated our bibliography to include some more recent citations, including those listed below:

use? Or if not, maybe it could be elaborated in your introduction more clearly why this study is so relevant.	Hennessy M, Hunter A, Healy P, et al. Improving trial recruitment processes: how qualitative methodologies can be used to address the top 10 research priorities identified within the PRioRiTy study. Trials 2018;19(1):1-5. Jennings CG, MacDonald TM, Wei L, et al. Does offering an incentive payment improve recruitment to clinical trials and increase the proportion of socially deprived and elderly participants? Trials 2015;16(1):1-9. Phelps EE, Tutton E, Griffin X, et al. A mixed-methods systematic review of patients' experience of being invited to participate in surgical randomised controlled trials. Social science & medicine 2020;253:112961. Stahl NA, King JR. Expanding approaches for research: Understanding and using trustworthiness in qualitative research. Journal of Developmental Education 2020;44(1):26-28. Freedman B. Equipoise and the ethics of clinical research. Human Experimentation and Research. Routledge;2017;427-431. Saunders B, Sim J, Kingstone T, et al. Saturation in qualitative research: exploring its conceptualization and operationalization. Quality & quantity 2018;52(4):1893-1907. Que J, Garman KS, Souza RF, et al. Pathogenesis and cells of origin of Barrett's esophagus. Gastroenterology 2019;157(2):349-364. e1.
---	--

VERSION 2 – REVIEW

REVIEWER	Sijben, Jasmijn Radboudumc Radboud Institute for Health Sciences
REVIEW RETURNED	22-Nov-2022
GENERAL COMMENTS	The authors have satisfactorily addressed my comments.
REVIEWER	van Driessche, Anne

	Vrije Universiteit Brussel Faculteit Geneeskunde en Farmacie
REVIEW RETURNED	21-Nov-2022

GENERAL COMMENTS	The added information strengthened this article, thank you for this important work!
---